# Antibiotic Use in Horses: Analysis of 57 German Veterinary Practices (2018–2023)

**DOI:** 10.3390/antibiotics14090953

**Published:** 2025-09-19

**Authors:** Roswitha Merle, Leonie Feuer, Katharina Frenzer, Jan-Lukas Plenio, Astrid Bethe, Nunzio Sarnino, Antina Lübke-Becker, Wolfgang Bäumer

**Affiliations:** 1Institute of Veterinary Epidemiology and Biostatistics, Veterinary Centre for Resistance Research, Freie Universität Berlin, 14163 Berlin, Germany; katharina.frenzer@fu-berlin.de (K.F.); jan-lukas.plenio@fu-berlin.de (J.-L.P.); nunzio.sarnino@fu-berlin.de (N.S.); 2Institute of Pharmacology and Toxicology, Freie Universität Berlin, 14195 Berlin, Germany; leonie.feuer@fu-berlin.de (L.F.); wolfgang.baeumer@fu-berlin.de (W.B.); 3Institute of Microbiology and Epizootics, Veterinary Centre for Resistance Research, Freie Universität Berlin, 14163 Berlin, Germany; astrid.bethe@fu-berlin.de (A.B.); antina.luebke-becker@fu-berlin.de (A.L.-B.); 4German Environment Agency, Wörlitzer Platz 1, 06844 Dessau-Roßlau, Germany

**Keywords:** antimicrobials, consumption, critically important antimicrobials, AMU, antibiotics monitoring, equine medicine

## Abstract

**Background/Objectives**: A mandatory monitoring of the use of antibiotics in horses in the European Union will come into force from 2027 on. The aim of the study was to explore the potential implementation of a monitoring system and to provide an overview of antibiotic use in horses in Germany. **Methods**: Data on all consultations from 57 German practices between 2018 and 2023 were obtained. The dataset included basic data about the horse, free-text diagnoses (allocated to one of 20 categories), and treatments. Information on the administered or dispensed pharmaceutical product was recorded for antibiotic treatment consultations. **Results**: This study analyzed 225,622 consultations with more than 50,000 horses. Antibiotics were administered in around 7% of consultations, but practice-specific rates varied considerably. Treatment was most frequent in ophthalmology cases. The most commonly used drug classes were sulfonamides combined with trimethoprim and aminopenicillins. Horses receiving antibiotics required follow-up visits more often than untreated animals, and changes in antibiotic substance occurred occasionally. **Conclusions**: Routine practice data provide valuable insights into antibiotic use in equine medicine. While incomplete entries and imprecise details (e.g., missing concentrations or diagnoses) remain a limitation, the approach offers clear advantages: it is cost-effective, allows large-scale data collection, and supports continuous monitoring over time. Such systems can be used to evaluate the effects of upcoming EU regulations and to identify priorities for antibiotic stewardship in equine practice.

## 1. Introduction

The use of antibiotics is a key factor in the selection of resistant bacteria [1]. In veterinary medicine, many antimicrobial substances used in human medicine are also administered to animals [2]. As a result, antibiotic use in animals may contribute to the emergence of resistant bacterial infections in humans, which can be challenging to treat [3]. To address this risk, the World Health Organization (WHO) has identified a list of critically important antimicrobials for human health and recommended restricting their use to human medicine [4]. In addition, the European Medicines Agency lists antimicrobial substances into categories A (Avoid) to D (Prudence) “based on the potential consequences to public health” [5].

The use of antibiotics in horses has rarely been subject to research so far. Schnepf et al. reported the results of a German teaching hospital in 2017 [6]. A total of 38.6% of the horses presented received antibiotics, mainly sulfonamides with trimethoprim (29.8%), aminoglycosides (14.7%), and penicillins (14.5%). A study in the UK revealed that 11.1% of 252 horses in 17 practices received antibiotic treatment [7]. Again, sulfonamides were the most commonly used substances (43.6%), followed by tetracyclines (23.2%) and penicillins. Mair and Parkin revealed a 50% reduction in antibiotic use in 11 UK equine practices from 2014 to 2018 [8]. The use of antibiotics, including critically important antibiotics—namely fluoroquinolones and cephalosporins—in horses at a French veterinary teaching hospital decreased between 2014 and 2020 after implementing French regulations on antibiotic use, but remained at a significantly higher percentage than in the other studies [9].

Antibiotic use in horses is also a relevant issue in the One Health context. Recent work has highlighted the significance of antimicrobial resistance in horses [10] and the value of routine data collection in the equine practice [6]. In addition, horses have been explicitly identified as part of One Health by Lönker et al., who underscored the need to include them in surveillance and stewardship initiatives [11].

So far, regular monitoring of antibiotic use in horses is not mandatory in the European Union, but it will come into force in 2027, when all member states will need to not only report sales data but also data on the use of antibiotics per animal species and antimicrobial substances [12].

To explore the potential implementation of a monitoring system in Germany, a feasibility study was conducted on behalf of the German Ministry of Food and Agriculture (support code 2820HS002). The data collected in this study were analyzed with the following objectives: (i) to provide an overview of antibiotic use in horses, both overall and at the practice level, (ii) to examine usage patterns by substance class, including the highest-priority critically important antibiotics, (iii) to assess antibiotic use by medical indication, and (iv) to analyze usage trends by age of horses.

## 2. Results

A total of 225,622 consultations involving 51,307 individual horses from 57 veterinary practices were analyzed. Practices were distributed all over Germany with a certain emphasis in the Northwest of Germany (Figure 1). The number of patients treated per practice ranged from 101 to 6419, with a median of 430 per practice. During a single consultation, one or more antibiotics could be administered or dispensed—for example, an injection given during the visit alongside a prescription for oral antibiotics for continued treatment at home. Practices had between 186 and 19,680 consultations with a median of 1720. Overall, antibiotics were used in 15,709 consultations (7.0%, 95% confidence interval (CI) 6.9–7.1%), with multiple antibiotics administered or dispensed in 1651 cases (10.5%, 95% CI 10.0–11.0%).

Practice-specific rates ranged from 0.0% to 17.0%, with an adjusted average of 6.5% (95% CI 6.4–6.7%, Figure 2). Ten veterinary practices did not prescribe or administer antibiotics to horses during the study period. Seven of them treated horses among other species, seven did not treat animals at all with antibiotics, while three practices treated cats and dogs, but not horses. Typical documentations were “acupuncture”, “osteopathics” or “adaptation of the saddle”, although other records covered clearly veterinary tasks such as blood sampling or vaccination. The comparison over the years showed only a slight downward trend from 5.1% in 2019 to 3.6% in 2023 (Appendix A Figure A1).

Of all antibiotic treatments, 73.8% were classified as initial treatments (no antibiotic treatment in the last 30 days). In 33.4% of these cases, a follow-up consultation involving antibiotic treatment occurred within seven days. Additionally, 4.4% of all horses received antibiotic treatment on at least two occasions within one year (at least 30 days between the treatments).

Horses that received antibiotic treatment tended to return earlier than those without treatment, e.g., within the first two days (47.2% of treated, 28.0% of non-treated horses). However, after day three, the return rate was higher among animals that had not been treated with antibiotics (Figure 3).

Of the 15,709 consultations involving antibiotic treatment, human pharmaceuticals were used in 1546 cases (9.8%, 95% CI 9.4–10.3%). All except one of them were administered as eye drops or ointments. Gentamicin was the most frequently used active ingredient (63.8%), followed by neomycin (18.4%) and ofloxacin (15.8%).

Topical administration accounted for 27.6% of all treatments, oral administration for 34.9%, and parenteral administration for 37.5%.

### 2.1. Use of Antibiotics by Substance Class

Among antibiotic substance classes, sulfonamides and trimethoprim were the most commonly used in horses, followed by aminopenicillins, which included amoxicillin, ampicillin, and benzylpenicillin. Amoxicillin without clavulanic acid accounted for 9.6% of all treatments, while amoxicillin with clavulanic acid was used in 0.72% of cases. Highest-Priority Critically Important Antimicrobials (HPCIAs) as classified by the WHO [4] were used in 8.5% of treatments (Figure 4).

The following substances were used in horses in our study, although they are not authorized for food-producing horses: cefovecin (n = 2), chloramphenicol (n = 101), clindamycin (n = 1), fusidic acid (n = 10), metronidazole (n = 3), ofloxacin (n = 245), and polymyxin B (n = 19, six-month withdrawal period), i.e., 2.2% of all consultations in total.

In some cases, multiple antibiotic pharmaceuticals were administered during a single consultation. This occurred in 6861 consultations (43.7% of all consultations with antibiotic treatment, 6497 fixed-dose combinations, 332 = 4.8% empirical combinations). Of the fixed-dose combinations, 99.9% were sulfonamides with trimethoprim, 4.4% (n = 302) were neomycin with polymyxin B, and 2.0% (n = 137) were amoxicillin with clavulanic acid. Among the empirical combinations, three pharmaceuticals were used in four cases, and four pharmaceuticals in two cases. The most common combination involved gentamicin and penicillin, which were prescribed in 93 consultations (28.0% of cases with empirical combinations). Neomycin and penicillin were combined in 23 consultations.

During follow-up visits, a change in the antibiotic substance occurred in 282 cases (8.8% of 3196 follow-up visits within seven days). In 34% of these cases, the switch happened on the first day after the initial treatment, and in 51% of cases, within the first two days. The most frequent changes involved switching from sulfonamides to penicillins (25 cases) or vice versa (49 cases).

### 2.2. Use of Antibiotics by Indication

Of all consultations, 74,326 (32.4%) were assigned to a specific indication group. The distribution of indications recorded during veterinary consultations is shown in Figure 5 and in Table 1. The most frequently assigned categories included diagnostics/therapy, dermatology, orthopedics, and respiratory diseases.

The percentage of consultations with antibiotic treatment varied between the indication groups, with a total of 9.7% consultations with antibiotic treatment and assignment to an indication group. Cases of ophthalmology were treated most frequently with antibiotics (53.9%, Figure 6, Table 1).

Typical diagnoses that were treated in the context of dermatology were abscesses, phlegmones, and dermatitis, and in 44.7% of cases, sulfonamides were used. Concerning ophthalmology, conjunctivitis or other infectious processes were the predominant diagnoses, with high usage of aminoglycosides (53.6%) and fluoroquinolones (9.7%), while coughing and even asthma were frequent diagnoses of the respiratory system (mostly sulfonamides (63.2%) or beta-lactams (18.9%) (Table 2). Ophthalmologic (93.4% of all ophthalmologic cases), otologic (50.0%), and diseases of the immune system (40.7%) were mainly treated with topical preparations, systemic (63.4%) and cardiologic diseases (61.5%) were treated parenterally, and for intoxications (71.4%), endocrinologic (68.2%) as well as respiratory problems (60.3%), oral treatment was preferred.

### 2.3. Use of Antibiotics by Horse’s Age

Horses were aged between 1 day and 39.9 years (animals above 40 years were excluded), with a median of 13.1 years and IQR 7.8–19.4 years. The lowest proportion of consultations involving antibiotic treatment was observed in horses aged between 5 and 20 years (Figure 7). In contrast, young horses up to three years old received significantly more antibiotic treatments, primarily due to trauma, respiratory diseases, and dermatology.

## 3. Discussion

As far as we know, this is the largest study on antibiotic use concerning the number of veterinary practices and consultations that has been published in Germany. Horses were treated in around 7% of consultations. Schnepf et al. found higher treatment rates in the teaching hospital in Hannover (38.7% of horses received antibiotic treatment) [6]. In another hospital study in Italy, the authors reported antimicrobial prescriptions in 42% of 1014 visits between 2011 and 2021. Hospitals usually use more antibiotics than outpatient practices due to a different spectrum of indications [13]. The higher percentage of parenteral applications by Bacci et al. [13] might also be since the study reports on data from a clinic and not from practices. Sinclair et al., as well as Allen et al., also reported slightly higher values in antibiotic treatments, although their studies derived from practices, not clinics [7,14]. It remains unclear why the practices in our study seem to have used fewer antibiotics than other studies reported. Since Lower Saxony, North Rhine-Westphalia, and Bavaria are the federal states with the highest number of horses [15] and also those with the highest number of participating practices in our study, we assume that the geographical distribution of our practices is rather representative.

At the same time, we observed a large variety between practices, including ten practices that never used antibiotics at all, although we carefully excluded non-veterinary practices, such as naturopaths and physiotherapists. These practices were rather small, with between 109 and 382 patients compared to more than 2000 patients in large practices. According to the records, these practices combined veterinary services with other non-veterinary treatments, and we do not know if the horse owners had a second veterinarian who eventually dispensed antibiotics in the study period. We were unable to distinguish between hospitals and practices, nor between first-, second-, and third-line practices, which is a limitation of our study. This information would be valuable for future research.

It is understandable why follow-up consultations tended to occur earlier in animals treated with antibiotics. From our experience, non-infectious diseases often require mid- to long-term treatment before their effectiveness becomes apparent, whereas the effects of antibiotic treatments are typically noticeable within a few days. If an antibiotic proves ineffective, owners are more likely to return to the veterinarian sooner for a follow-up consultation. Some animals were re-presented to the veterinarian as early as the next day—an occurrence that was more common among treated animals than non-treated ones.

Bacci et al. [13] also saw the frequent use of combinations of substances, mainly due to the combination of sulfonamides with trimethoprim (87% of all combined therapies). A combination of penicillin and gentamicin is one of the most commonly used drugs for perioperative antibiotic prophylaxis in equine surgery, including abdominal surgery (colic) [16,17,18].

It is known that young horses are more prone to infectious diseases as their immune system is not yet fully developed [19]. We could show that 10- to 15-year-old horses had the lowest percentage of consultations with antibiotic treatment. But this is rather because typical diseases of adult horses are in the musculoskeletal system (lameness) or the digestive system (colic), which do not need antibiotic therapy. To our knowledge, no other study has reported the effect of age on the need for antibiotic treatment.

In our study, nearly 10% of prescriptions involved pharmaceuticals registered for human use. Such prescriptions may reflect shortages of veterinary medicines for specific therapeutic needs in the veterinary practice, but they also raise concerns regarding antimicrobial resistance and compliance. From a One Health perspective, the off-label use of human antibiotics in animals highlights the importance of prudent use and the need for continued monitoring to minimize potential risks for both animal and public health [10].

Human pharmaceuticals were almost exclusively eye drops or ointments. Obviously, no veterinary product containing cortisone and gentamicin is authorized in Germany. The percentage of treatments with human pharmaceuticals was addressed by Schnepf et al. They reported that, apart from gentamicin, many penicillins licensed for humans were used [6]. Bacci et al. reported that 14% of prescriptions of systemic antimicrobials were off-label use of human products [13].

In addition, the use of substances that are not authorized for food-producing horses is a concern that may not be neglected. In Germany, horses must actively opt out of food production [12,20]. In 2021, a total of 3489 horses were slaughtered for human consumption in Germany, which, given an overall horse population of nearly one million, indicates a negligible proportion (<1%) [21]. We did not know which of the animals in our study were intended for food production, but we assume that this problem is negligible.

In around two-thirds of the treatments, sulfonamides with trimethoprim or aminopenicillins were used. Other studies also reported that sulfonamides are the most frequently used substance class in horses [6,7,8]. Sulfonamides combined with trimethoprim are popular because they have a broad effect, can be administered via feed, and are not restricted due to their rare use in humans [22]. The results of the use of amoxicillin as penicillin correspond to those of Schnepf et al., who found that around 12% of all applications were amoxicillin in Germany [6]. This class seems to be less popular in other countries since publications from the UK and Australia [23] do not mention it specifically, but report on the frequent use of penicillins and procaine penicillin (e.g., 5–6% in the UK [8], 16% in the UK [7]). Substances not authorized for use in food-producing horses in Germany were used in only 2.2% of all treatments. Since we did not know how many horses were de-registered as food-producing, we cannot assess this percentage further. There is also no publicly available information on how many horses are registered for slaughter in Germany.

The German Regulation of Veterinary Pharmacies seeks to encourage the responsible use of antimicrobials [24]. Antimicrobial susceptibility testing is mandatory before prescribing third- and fourth-generation cephalosporins, colistin, or fluoroquinolones. As a result, many veterinarians in Germany initially opt for amoxicillin as a first-line treatment and switch to an alternative if necessary.

The use of HPCIA treatments—especially of fluoroquinolones—was higher than reported by Schnepf et al. in Germany (0.9% of applications) [6], between less than 1% and 4% of prescriptions in the UK [7,25], similar to the USA and Canada (7.5%) [26], but less than in Italy (24%) [13] and Rule et al. [27] in the USA (25.7%), which might reflect that the data from these studies are older. Mair and Parkin reported a 38% reduction in enrofloxacin use, together with an almost complete reduction in the use of 3rd and 4th generation cephalosporins between 2014 and 2018 in the UK [8]. Also, another study reported on a decrease in the total amount of active ingredients over a ten-year period [28]. However, our study did not observe a significant reduction in overall antibiotic use over the years (see Appendix A Figure A1).

Since dermatologic diseases such as summer eczema or allergic dermatitis are very common among horses, and those horses need constant therapy [29], the high number of consultations with the indication of dermatology was not surprising. Also, orthopedic problems such as lameness and diseases of the respiratory system, i.e., equine asthma, occur often in horses. They need therapy, but not typically antibiotic treatment. This might contribute to the lower prevalence of antibiotic treatment in horses compared to other companion animals, such as dogs and cats [30].

A high percentage of consultations involving antibiotic treatment were observed for indications commonly associated with bacterial infections, mainly ophthalmology, but also trauma and respiratory diseases. However, generic indication categories such as emergency and systemic diseases also revealed high percentages of antibiotic treatments, which cannot be easily explained without more specific diagnoses. Our figures roughly correspond to the findings of Sinclair et al., Rule et al., and Bacci et al. [7,13,27]. Topical treatments were frequent in the indications “ophthalmology” and “otology”. Surprisingly, only a small part of the treatments concerning the generic group “dermatology” or the urogenital tract were treated topically.

Unfortunately, we were only able to assign a supposed indication in 32% of all consultations, and even these assignments proved questionable upon random manual inspection: 22.7% (95% CI 18.3–27.7%) of 300 consultations were incorrectly allocated. The 95% CI indicates that even 27.7% of the data might be wrongly allocated. One very promising approach from the UK is PetBERT, a domain-specific language model based on the BERT architecture. It has been trained on veterinary clinical records from 253 small animal practices to detect outbreaks [31]. PetBERT is designed to capture the terminology and context of veterinary medicine, enabling more accurate tasks such as diagnosis classification, entity recognition, and monitoring of treatments. In the future, advancements in artificial intelligence and text mining methods are expected to improve accuracy and yield more precise results.

Another limitation of this study is the use of routine data, which included missing information in some entries (e.g., lack of anamnesis or recorded diagnosis) and imprecise documentation (e.g., “amoxicillin” listed as the drug name without specifying the trade name or concentration). More detailed information on diagnoses, the exact trade name of the pharmaceutical, or the duration of treatment would have allowed for a more in-depth assessment of the reasons behind antibiotic use. We also had no information concerning consultations of individual horses in other practices. However, one of our objectives was to assess how useful routine data can be for monitoring purposes.

In conclusion, analyzing routine data from 57 veterinary practices over six years in Germany enabled differentiation between substances and the categorization of treatments by indication. As a result, it becomes possible to monitor changes over time and to assess the impact of legislative regulations.

While routine data collection has some limitations due to missing specific details, its accessibility at a low cost often outweighs these drawbacks. Additionally, since veterinarians record treatments for invoicing purposes, the data remains comprehensive and largely unbiased. Thus, the use of such data for passive monitoring is feasible after some adaptations. Importantly, integrating antibiotic use contributes to a broader One Health perspective by addressing the interconnectedness of animal, human, and environmental health. In this context, recording of the treatment duration as well as the exact trade name, including the concentration of the active ingredients, is indispensable for monitoring systems.

## 4. Materials and Methods

The study received ethical approval from the Ethics Commission of Freie Universität Berlin, Germany (Approval Number: ZEA-Nr. 2021-018).

### 4.1. Data Collection and Processing

Data on antibiotic usage were extracted from debevet, a cloud-based veterinary practice management system headquartered in Berlin, Germany. The dataset encompassed an anonymized identifier for each veterinary practice and included essential patient details such as species, breed, sex, neuter status, date of birth, and, where available, date of death. Additionally, each consultation was recorded with a corresponding date and free-text entries for anamnesis and diagnoses.

Each record within the dataset corresponded to a single invoiced item, which could represent a medical procedure, diagnostic test, pharmaceutical product, or other billed services. For analytical purposes, all entries from the same day and referring to the same horse were consolidated into a single consultation event. Consultations could involve antibiotic treatment or not, and any instance where an antibiotic was either administered by the veterinarian or dispensed to the owner was classified as treatment. Throughout this study, “treatment” specifically refers to antibiotic usage.

### 4.2. Classification of Diagnoses and Treatments

All free-text information associated with each case—anamnesis, diagnoses, and case descriptions—was concatenated into a single string and screened against a veterinary keyword catalog. The catalog was assembled from published nomenclatures (Gohrbandt [32] and Allenspach et al. [33]), three institutional diagnostic lists, a table of differential diagnoses, and the most frequent words occurring in initially unmatched records; every term was reviewed by two veterinarians and assigned to one of 20 diagnostic categories (see Table 1). Successive expansion yielded ~24,000 unique terms covering diagnoses, procedures, and anatomical sites. Automated matching was carried out in Python 3.11 (Python Software Foundation, Wilmington, DE, USA) with the scikit-learn library (v1.6, [34]). CountVectorizer converted the case texts into a term–frequency matrix restricted to the catalog vocabulary; a case was labeled with every diagnostic category for which ≥1 keyword was detected, allowing multi-category assignments when appropriate.

Manual inspection of 300 consultations with allocated diagnostic categories revealed 68 incorrect allocations (22.7%). Typical mistakes were the allocation of a vaccination to the respective organ system instead of “Diagnostics/Therapy”, e.g., influenza vaccination was allocated to the respiratory system.

### 4.3. Classification of Antibiotics and Source of Pharmaceutical Data

For consultations involving antibiotic treatment, detailed information was available regarding the product administered or dispensed, including:Quantity per package.Active substance(s) and concentrations.Route of administration.

To ensure consistency in classification, data on veterinary-specific pharmaceuticals were sourced from vetidata.de [35] (a database that details all authorized veterinary pharmaceuticals on the German market), while details on human medicinal products were retrieved from the “Rote Liste” database [36]. Antimicrobials classified as Highest-Priority Critically Important Antimicrobials, as defined by the WHO, were specifically noted [4]. These include 3rd- and 4th-generation cephalosporins, fluoroquinolones, macrolides, and polymyxins.

### 4.4. Inclusion Criteria and Study Period

Only data from veterinary practices that had at least 100 registered equine patients during the study period were included in the analysis. The dataset spanned 2018 to 2023, as adoption of debevet (available since 2014) was limited before 2018.

### 4.5. Statistical Analyses

Data processing and statistical analysis were performed in RStudio (Version 2024.04.2) using R (version 4.4.1), leveraging packages such as “tidyverse,” “here,” “ggplot2,” “mgcv”, and “scales” [37,38,39,40,41,42]. We calculated percentages, including 95% confidence intervals (95% CI), using the Wilson approximation. To compare the percentages over the years 2019–2023, only those 34 practices were included that provided data over all years. To calculate prevalence and 95% adjusted for the practice effect, a common effects meta-analysis was conducted using the R package “meta”, assuming that all studies estimated the same underlying true effect and weighted studies according to the inverse of their within-study variance [43].

### 4.6. Mapping

We obtained geospatial data for 5-digit German postal code areas from the Opendatasoft public dataset (georef-germany-postleitzahl). The data were imported and processed in R using the sf [44], dplyr, and ggplot2 packages. Postal codes were truncated to the first two digits to represent aggregated postal regions, and corresponding geometries were merged using spatial union operations. Aggregated counts of veterinary practices per 2-digit postal code region were merged with the geospatial data using a left join on the postal code prefix. The resulting map was visualized using a monochromatic gradient (“Blues”) with scale_fill_distiller() to reflect the number of veterinary practices per region.

Initial treatments were defined as consultations where no antibiotic treatment had been recorded for the preceding 30 days. Follow-up treatments referred to cases where an antibiotic was prescribed again within seven days of an initial treatment. To assess antibiotic switches during follow-up visits, only cases where a single antibiotic was used per consultation were analyzed, resulting in 11,590 consultations classified as initial treatments. In addition, the time to follow-up consultation was analyzed for horses with and without treatment, including follow-up consultations between 1 and 21 days after the initial consultation (80,013 follow-up consultations, thereof 7077 with and 72,936 without treatment).

To explore the impact of the horse’s age on antibiotic prescription rates, a non-linear logistic regression model incorporating natural splines (three degrees of freedom) was applied after removing implausible values <0 and >40 years (141,091 consultations; excluded due to missing age: 84,531). The 95% confidence intervals for this model are presented in Figure 6.

We used ChatGPT 5 (OpenAI, San Francisco, CA, USA; accessed January to August 2025) to assist in troubleshooting and refining R code during the data analysis. All code and results were independently verified by the authors.

## Figures and Tables

**Figure 1 antibiotics-14-00953-f001:**
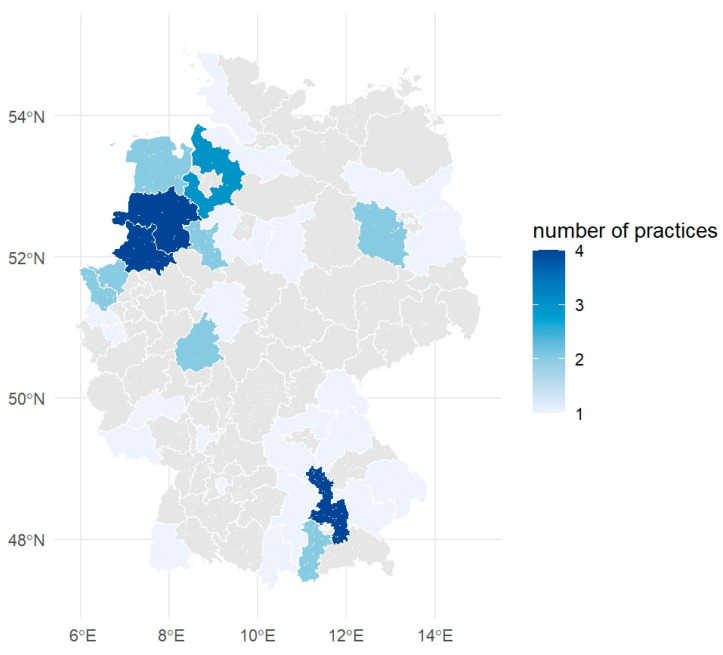
Geographical distribution of 57 participating horse practices.

**Figure 2 antibiotics-14-00953-f002:**
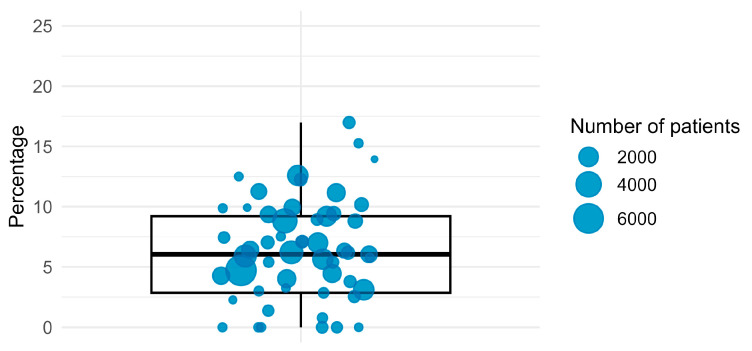
Percentage of consultations with antibiotic treatment per veterinary practice; n = 225,622 consultations from 57 German horse practices from 2018 to 2023.

**Figure 3 antibiotics-14-00953-f003:**
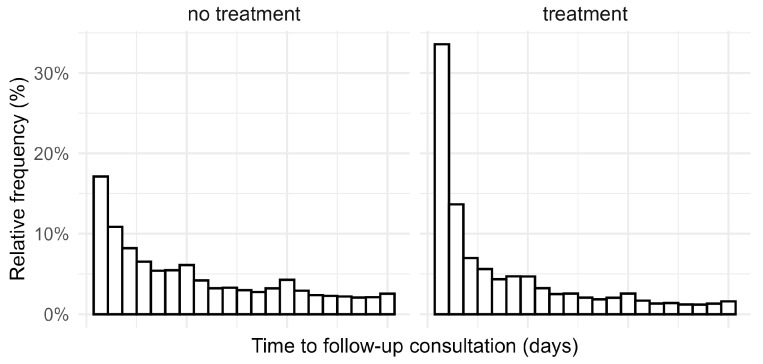
Histogram of time to follow-up consultations of horses, categorized by consultations with or without treatment with antibiotics.

**Figure 4 antibiotics-14-00953-f004:**
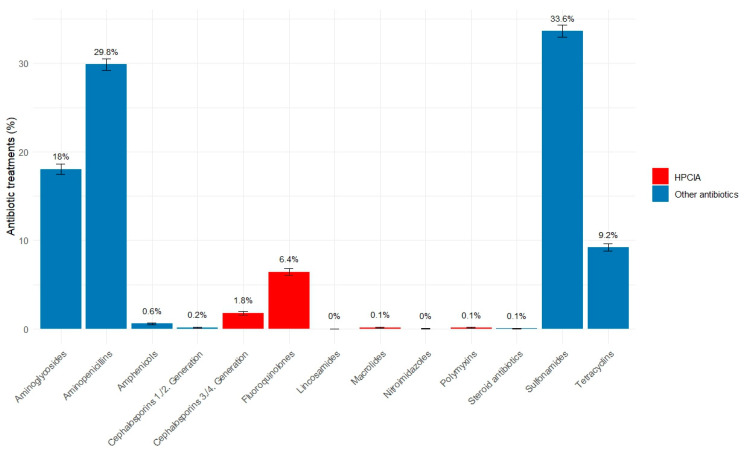
Percentage including 95% confidence intervals of consultations with antibiotic treatment per substance class in horses (17,633 applications or deliveries from 57 German horse practices from 2018 to 2023). Highlighted are substance groups that belong to the highest-priority critically important antibiotics.

**Figure 5 antibiotics-14-00953-f005:**
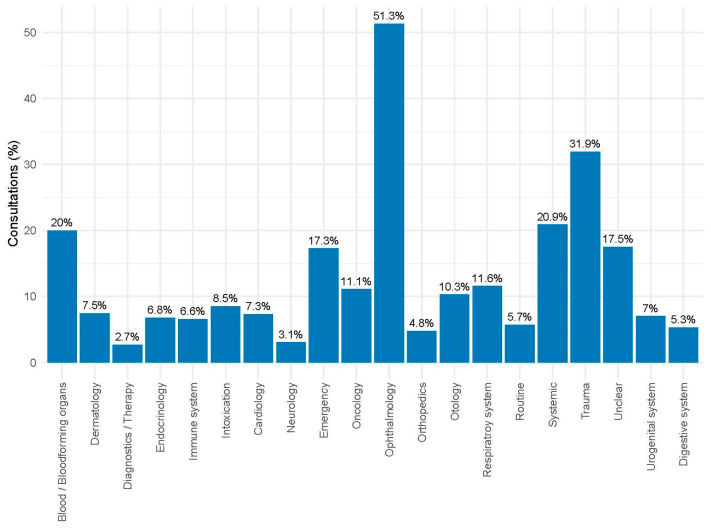
Percentage of consultations per indication group in 74,326 consultations of horses from 57 German horse practices from 2018 to 2023.

**Figure 6 antibiotics-14-00953-f006:**
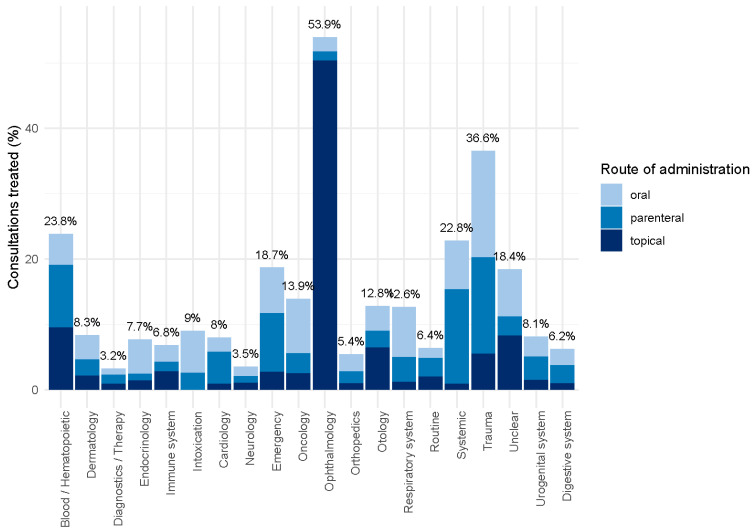
Percentage of consultations with antibiotic treatment per indication group in 7276 treatments of horses from 57 German horse practices from 2018 to 2023.

**Figure 7 antibiotics-14-00953-f007:**
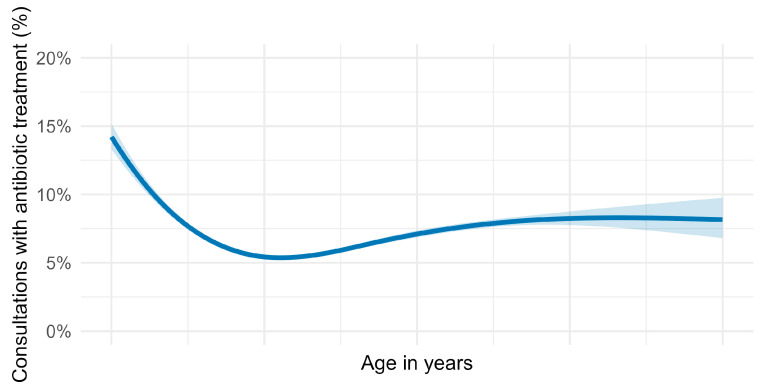
Percentage of visits with antibiotic treatment in horses over age. Displayed are the estimates for a non-linear regression model with natural splines, 3 degrees of freedom, and a 95% confidence interval, n = 141,091 consultations from 57 German horse practices from 2018 to 2023.

**Table 1 antibiotics-14-00953-t001:** Number and percentage of consultations, percentage of consultations treated, and route of administration per indication; 75,173 consultations of 57 horse practices from 2018 to 2023.

Indication	No. of Consultations	% of All Consultations with 95% CI	% Treated (Total) with 95% CI	Oral (%)	Parenteral (%)	Topical (%)
Blood/Hematopoietic organs	20	0% (0–0%)	23.8% (14.2–37.2%)	20.0	40.0	40.0
Dermatology	13,761	18.5% (18.2–18.8%)	8.3% (7.9–8.7%)	44.5	29.8	25.8
Diagnostics/Therapy ^1^	17,302	23.3% (23.0–23.6%)	3.2% (2.9–3.4%)	29.7	43.5	26.8
Endocrinology ^2^	570	0.8% (0.7–0.8%)	7.7% (5.6–10.4%)	68.2	13.6	18.2
Immune system	396	0.5% (0.5–0.6%)	6.8% (4.6–9.9%)	37.0	22.2	40.7
Intoxication	78	0.1% (0.1–0.1%)	9.0% (4.1–17.5%)	71.4	28.6	0.0
Cardiology	1354	1.8% (1.7–1.9%)	8.0% (6.6–9.6%)	27.5	61.5	11.0
Neurology	2205	3% (2.8–3.1%)	3.5% (2.8–4.3%)	40.8	30.3	28.9
Emergency	548	0.7% (0.7–0.8%)	18.7% (15.6–22.3%)	37.5	48.1	14.4
Oncology	469	0.6% (0.6–0.7%)	13.9% (10.9–17.4%)	59.7	22.4	17.9
Ophthalmology	2192	2.9% (2.8–3.1%)	53.9% (51.9–55.9%)	4.0	2.6	93.4
Orthopedics	8948	12% (11.8–12.3%)	5.4% (4.9–5.9%)	48.7	33.8	17.5
Otology	152	0.2% (0.2–0.2%)	12.8% (8.4–18.9%)	30.0	20.0	50.0
Respiratory system	7708	10.4% (10.2–10.6%)	12.6% (11.8–13.4%)	60.3	30.6	9.1
Routine exam/prevention ^3^	6667	9% (8.8–9.2%)	6.4% (5.8–7.0%)	24.6	44.3	31.1
Systemic disease ^4^	567	0.8% (0.7–0.8%)	22.8% (19.5–26.5%)	32.8	63.4	3.8
Trauma	3079	4.1% (4–4.3%)	36.6% (35.0–38.3%)	44.6	40.4	15.0
Unclear/unspecified ^5^	408	0.5% (0.5–0.6%)	18.4% (14.8–22.6%)	39.5	15.8	44.7
Urogenital system	4564	6.1% (6–6.3%)	8.1% (7.3–8.8%)	37.6	44.5	17.9
Digestive system	3338	4.5% (4.3–4.6%)	6.2% (5.5–6.9%)	40.2	44.5	15.3
Total	74,326	100.0%	9.7% (9.5–10.0%)	36.9	31.3	31.8

^1^ procedures such as vaccinations, castration, euthanasia, diagnostic exams, and preventive treatments. ^2^ e.g., thyroid disorders and PPID. ^3^ antiparasitic treatments, drug dispensing, and microchipping. ^4^ general symptoms such as lethargy, anorexia, and fever. ^5^ terms that lacked specific diagnostic value (e.g., “ultrasound” or “edema”).

**Table 2 antibiotics-14-00953-t002:** Antibiotic classes by indication; number (percentage) of prescriptions per substance class and indication.

Indication	Aminoglycosides	Beta-Lactams	Sulfonamides	Amphenicols	Cephalosporins (1st/2nd Gen)	Cephalosporins (3rd/4th Gen)	Fluoroquinolones	Polymyxins	Steroid Antibiotics	Tetracyclines	Macrolides	Nitroimidazoles
Blood/Hematopoietic organs	3 (60.0%)	1 (20.0%)	1 (20.0%)	-	-	-	-	-	-	-	-	-
Dermatology	222 (19.2%)	250 (21.6%)	517 (44.7%)	3 (0.3%)	3 (0.3%)	36 (3.1%)	59 (5.1%)	1 (0.1%)	2 (0.2%)	63 (5.4%)	-	-
Diagnostics/Therapy	123 (22.1%)	176 (31.7%)	167 (30.0%)	1 (0.2%)	-	14 (2.5%)	20 (3.6%)	2 (0.4%)	-	51 (9.2%)	2 (0.4%)	-
Endocrinology	7 (15.9%)	4 (9.1%)	24 (54.5%)	-	-	1 (2.3%)	7 (15.9%)	-	-	1 (2.3%)	-	-
Immune system	9 (33.3%)	5 (18.5%)	10 (37.0%)	-	-	-	1 (3.7%)	-	-	2 (7.4%)	-	-
Intoxication	-	1 (14.3%)	5 (71.4%)	-	-	-	1 (14.3%)	-	-	-	-	-
Cardiology	10 (9.2%)	48 (44.0%)	35 (32.1%)	-	1 (0.9%)	5 (4.6%)	6 (5.5%)	-	-	4 (3.7%)	-	-
Neurology	16 (21.1%)	15 (19.7%)	32 (42.1%)	-	-	1 (1.3%)	5 (6.6%)	-	-	7 (9.2%)	-	-
Emergency	11 (10.6%)	32 (30.8%)	43 (41.3%)	-	-	4 (3.8%)	8 (7.7%)	-	-	6 (5.8%)	-	-
Oncology	11 (16.4%)	10 (14.9%)	41 (61.2%)	-	-	-	1 (1.5%)	-	-	4 (6.0%)	-	-
Ophthalmology	662 (53.6%)	30 (2.4%)	51 (4.1%)	39 (3.2%)	-	2 (0.2%)	112 (9.1%)	-	2 (0.2%)	337 (27.3%)	-	-
Orthopedics	59 (12.2%)	124 (25.6%)	236 (48.7%)	1 (0.2%)	-	6 (1.2%)	16 (3.3%)	-	-	43 (8.9%)	-	-
Otology	7 (35.0%)	3 (15.0%)	5 (25.0%)	-	-	-	3 (15.0%)	2 (10.0%)	-	-	-	-
Respiratory system	54 (5.5%)	185 (18.9%)	618 (63.2%)	4 (0.4%)	1 (0.1%)	18 (1.8%)	57 (5.8%)	-	-	40 (4.1%)	1 (0.1%)	-
Routine exam/prevention	86 (20.7%)	174 (41.9%)	96 (23.1%)	-	2 (0.5%)	-	16 (3.9%)	1 (0.2%)	-	36 (8.7%)	4 (1.0%)	-
Systemic disease	6 (4.6%)	66 (50.4%)	50 (38.2%)	-	-	-	7 (5.3%)	-	-	2 (1.5%)	-	-
Trauma	119 (9.9%)	398 (33.1%)	561 (46.7%)	1 (0.1%)	1 (0.1%)	23 (1.9%)	36 (3.0%)	-	-	62 (5.2%)	-	-
Unclear/unspecified	13 (17.1%)	9 (11.8%)	26 (34.2%)	2 (2.6%)	-	-	13 (17.1%)	-	-	13 (17.1%)	-	-
Urogenital system	47 (12.5%)	117 (31.2%)	146 (38.9%)	-	1 (0.3%)	14 (3.7%)	21 (5.6%)	-	-	29 (7.7%)	-	-
Digestive system	29 (13.9%)	60 (28.7%)	91 (43.5%)	-	-	2 (1.0%)	14 (6.7%)	-	-	12 (5.7%)	-	1 (0.5%)

## Data Availability

The data presented in this study are available on request from the corresponding author. The data are not publicly available due to privacy restrictions.

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
