# Peer review of "Antibiotic Use in Horses: Analysis of 57 German Veterinary Practices (2018–2023)"

_antibiotics, 2025, doi:10.3390/antibiotics14090953_

Round 1

Reviewer 1 Report

Comments and Suggestions for Authors

The study covers more than 225000 consultations and over 51000 horses across 57 veterinary practices, making it the largest investigation on equine antibiotic use in Germany. The research is relevant in the context of antimicrobial resistance (AMR) surveillance efforts and is directly connected to upcoming regulatory changes in the EU, which underscores its high practical significance.

Comments:
- 1st page, bottom left: edit data to fit the new article, not the published article “Use of Antibiotics in Companion Animals from 133 German Practices from 2018 to 2023”.
- The abstract is overloaded with details (exact percentages, confidence intervals, enumerations), which reduces readability. It should be shortened by limiting quantitative data and focusing on the main findings and practical implications.
- Line 123: In Figure 4 “Error! Reference source not found” is displayed.
-  Line 191: “Figure 6” should be labeled as “Figure 7”.
-  Line 293: “PetBERT” – a brief explanation of how PetBERT works would help readers unfamiliar with it.
-  Line 364: The study period (2018-2023) is mentioned, but the rationale for choosing this interval is not explained.
-  The sample of clinics emphasizes northwest Germany (lines 75-76), which may limit the representativeness of the data for the entire country. This factor should be highlighted separately in the Limitations section.
-  The authors report that 9.8% of prescriptions were for drugs registered for human use (lines 111-112). A more detailed analysis of the reasons for such use (shortage of veterinary drugs, specific indications, etc.) and the potential risks in terms of AMR and the One Health policy should be provided in the Discussion.
Recommendations:
-  The study aim is formulated but not clearly separated from the general overview. It is recommended to present it as a separate short sentence: “The aim of this study was to evaluate …”.
- Lines 210-211: Exclude clinics where antibiotics were not used at all.
-  Although the reference includes recent works (2020-2025), it would be beneficial to supplement the review with publications addressing the One Health concept.
-  The situation with food-producing horses remains unclear (lines 129-132). Since the use of certain antibiotics is prohibited in such animals, it is necessary, if possible, to analyze the impact of this variable. Moreover, the issue of legal consequences and control measures in such cases is not addressed.
Conclusion:
The article is suitable for publication after revision.

Author Response

- 1st page, bottom left: edit data to fit the new article, not the published article “Use of Antibiotics in Companion Animals from 133 German Practices from 2018 to 2023”.

Dear Reviewer, thank you for this hint. This footnote is adapted by MDPI’s editorial office.

- The abstract is overloaded with details (exact percentages, confidence intervals, enumerations), which reduces readability. It should be shortened by limiting quantitative data and focusing on the main findings and practical implications.

Thank you for this comment. We overworked the abstract accordingly (ll. 25-36):

Results: This study analyzed 225,622 consultations with more than 50,000 horses. Antibiotics were administered in around 7% of consultations, but practice-specific rates varied considerably. Treatment was most frequent in ophthalmology cases. The most commonly used drug classes were sulfonamides combined with trimethoprim and aminopenicillins. Horses receiving antibiotics required follow-up visits more often than untreated animals, and changes of antibiotic substance occurred occasionally. Conclusions: Routine practice data provide valuable insights into antibiotic use in equine medicine. While incomplete entries and imprecise details (e.g., missing concentrations or diagnoses) remain a limitation, the approach offers clear advantages: it is cost-effective, allows large-scale data collection, and supports continuous monitoring over time. Such systems can be used to evaluate the effects of upcoming EU regulations and to identify priorities for antibiotic stewardship in equine practice.”

- Line 123: In Figure 4 “Error! Reference source not found” is displayed.

Dear reviewer, please apologize for this mistake, we corrected it.

-  Line 191: “Figure 6” should be labeled as “Figure 7”.

Dear reviewer, we corrected that.

-  Line 293: “PetBERT” – a brief explanation of how PetBERT works would help readers unfamiliar with it.

Dear reviewer, sure, we added the following explanation in ll. 310-315: “One very promising approach from the UK is PetBERT, a domain-specific language model based on the BERT architecture. It has been trained on veterinary clinical records from 253 small animal practices to detect outbreaks [27]. PetBERT is designed to capture the terminology and context of veterinary medicine, enabling more accurate tasks such as diagnosis classification, entity recognition, and monitoring of treatments.“

-  Line 364: The study period (2018-2023) is mentioned, but the rationale for choosing this interval is not explained.

Dear reviewer, thank you for this comment. The debevet software came to market in 2014, and it took several years for a substantial number of practices to adopt this software. In 2018, 47 horse practices used debevet; in 2023, it was 145 practices. We received the data, including the 2023 data, in the summer of 2024.

We added in ll. 388-389: “The dataset spanned 2018 to 2023, as adoption of debevet (available since 2014) was limited before 2018.”

-  The sample of clinics emphasizes northwest Germany (lines 75-76), which may limit the representativeness of the data for the entire country. This factor should be highlighted separately in the Limitations section.

Dear reviewer, thank you for this comment. Actually, we believe that the geographical distribution is rather representative compared to the number of horses in the federal states. We mentioned this in ll. 212-215.

-  The authors report that 9.8% of prescriptions were for drugs registered for human use (lines 111-112). A more detailed analysis of the reasons for such use (shortage of veterinary drugs, specific indications, etc.) and the potential risks in terms of AMR and the One Health policy should be provided in the Discussion.

Dear reviewer, thank you for raising this important aspect. We added the following in ll. 244-249: “In our study, nearly 10% of prescriptions involved pharmaceuticals registered for human use. Such prescriptions may reflect shortages of veterinary medicines for spe-cific therapeutic needs in the veterinary practice, but they also raise concerns regarding antimicrobial resistance and compliance. From a One Health perspective, the off-label use of human antibiotics in animals highlights the importance of prudent use and the need for continued monitoring to minimize potential risks for both animal and public health [10].”

Recommendations:
-  The study aim is formulated but not clearly separated from the general overview. It is recommended to present it as a separate short sentence: “The aim of this study was to evaluate …”.

Thank you for this comment. We changed the second sentence as recommended to “The aim of the study was to explore the potential implementation of a monitoring system and to provide an overview of antibiotic use in horses in Germany.” (ll. 20-22)

- Lines 210-211: Exclude clinics where antibiotics were not used at all.

Dear reviewer, we understand your point and have discussed the pros and cons thoroughly. We believe it is important to leave these practices in the dataset, because we want to show the situation as it is, and this obviously includes practices that don’t use antimicrobials.

-  Although the reference includes recent works (2020-2025), it would be beneficial to supplement the review with publications addressing the One Health concept.

Dear reviewer, thank you very much for this valuable comment. We added a paragraph in the introduction in ll. 61-65:

“Antibiotic use in horses is also a relevant issue in the One Health context. Recent work has highlighted the significance of antimicrobial resistance in horses [10] and the value of routine data collection in the equine practice [6]. In addition, horses have been explicitly identified as part of One Health by Lönker et al., who underscored the need to include them in surveillance and stewardship initiatives [11].”

In addition, we also addressed this in the Conclusions in ll. 334-336:

“Importantly, integrating antibiotic use contributes to a broader One Health perspective by addressing the interconnectedness of animal, human, and environmental health.”

-  The situation with food-producing horses remains unclear (lines 129-132). Since the use of certain antibiotics is prohibited in such animals, it is necessary, if possible, to analyze the impact of this variable. Moreover, the issue of legal consequences and control measures in such cases is not addressed.

Dear reviewer, thank you for this point that should not be overlooked. We added in ll. 256-261 the following:

“In addition, the use of substances that are not authorized for food-producing horses is a concern that may not be neglected. In Germany, horses must actively opt out of food production [10, 18]. In 2021, a total of 3,489 horses were slaughtered for human consumption in Germany, which, given an overall horse population of nearly one million, indicates a negligible proportion (< 1%) [19]. We did not know which of the animals in our study were intended for food production, but we assume that this problem is negligible.”

Conclusion:
The article is suitable for publication after revision.

Dear reviewer, thank you very much for your comments, which really helped to improve the quality of the manuscript.

Reviewer 2 Report

Comments and Suggestions for Authors

The study by Roswitha Merle et al entitle ed “Antibiotic Use in Horses: Analysis of 57 German Veterinary Practices (2018–2023)” is aimed to the monitoring and analysis of antibiotic use in horses in Germany.  The study addressed important and relevant topic in the context of the upcoming EU-wide mandatory reporting in 2029 for the antibiotic uses in Horses. The manuscript is well-written, and the result is clearly stated, and the tables and figures are well presented. The discussion well structures and provide in-depth discussions the finding and highlights strengths and limitations of the study. The conclusion is consistent with the evidence presented in the results and properly addressing the study question. However, minor revision is needed before considering it for publication

-----------------------------------------------------------------------------------------------

Comment for authors

  1. Abstract: the keyword listed is good, but I suggest including “Antibiotics monitoring, horses or equine medicine “to the lists of keywords to increase the visibility   of your paper and describe the full scope the study.
  2. Introduction: in line 56: “Mair and Parkin revealed a 50% reduction in antibiotic use in 6 UK equine practices from 2014 to 2018 [8].”  When I check the actually cited source found the study included 11 UK equine practices not 6. Please revised
  3. “antimicrobials” and “Antibiotics” are used somewhat interchangeably across manuscript. please specify whether the cited sources refer to antimicrobials use or antibiotic use
  4. Line 57- 60: Please specify and provide detail information on the cases of Frech study. please include the specific name of antibiotic and year of study.
  5. Line 123: please remove the typos
  6. Figure 4 is not clear. please improve the figure resolution and quality.
  7. Please adjust column width in table 1 and 2 to improve readability.
  8. Line 187: figure 7 was cited in the main text but did not find in the manuscript. please kindly provide figure 7

Author Response

  1. Abstract: the keyword listed is good, but I suggest including “Antibiotics monitoring, horses or equine medicine “to the lists of keywords to increase the visibility   of your paper and describe the full scope the study.

Dear reviewer, thank you for this idea. We added antibiotics monitoring and equine medicine to the keywords.

  1. Introduction: in line 56: “Mair and Parkin revealed a 50% reduction in antibiotic use in 6 UK equine practices from 2014 to 2018 [8].”  When I check the actually cited source found the study included 11 UK equine practices not 6. Please revised

Dear reviewer, thank you for this comment, we corrected to 11 UK equine practices (l. 57)

  1. “antimicrobials” and “Antibiotics” are used somewhat interchangeably across manuscript. please specify whether the cited sources refer to antimicrobials use or antibiotic use

Dear reviewer, thank you. We harmonised on “antibiotics use”. In cases where we cited other authors or used established terms such as critically important antimicrobials, we did not adapt the text; thus, both terms still occur in the text.

  1. Line 57- 60: Please specify and provide detail information on the cases of Frech study. please include the specific name of antibiotic and year of study.

Dear author, we added years and substances to the sentence as follows in ll. 58-61: “The use of antibiotics, including critically important antibiotics – namely fluoroquinolones and cephalosporins – in horses at a French veterinary teaching hospital decreased between 2014 and 2020 after implementing French regulations on antibiotic use, but remained at a significantly higher percentage than in the other studies [9].“

  1. Line 123: please remove the typos

Dear reviewer, we apologize, but we cannot find any typos in l. 123.

  1. Figure 4 is not clear. please improve the figure resolution and quality.

Dear reviewer, we improved the figure resolution of all figures.

  1. Please adjust column width in table 1 and 2 to improve readability.

Dear reviewer, table 2 is hard to read in portrait orientation. We uploaded the table in addition in landscape format.

  1. Line 187: figure 7 was cited in the main text but did not find in the manuscript. please kindly provide figure 7.

Dear reviewer, we have adapted the numbering of the figures. The last figure is Figure 7 and is also referred to in the text.

Dear reviewer, thank you very much for your valuable comments and support. We did the best to overwork the manuscript to your convenience.

Reviewer 3 Report

Comments and Suggestions for Authors

Dear Authors

Line 334 'concentration'

It appears that you have made the very best use of the data available. The general use of antibiotics seems to be much more refined than was probably the case years ago. The veterinarians should be applauded in  restricting the use in most categories. Interpretation of the results was unfortunately limited by the data limitations. It would have been valuable to know why and under what circumstances the more valuable antibiotics were used. Also although the data was arranged into different categories, they were too general to examine the use of different antibiotics in specific disease areas. For instance prophilactic treatment after major abdominal surgery compared with a cutaneous tumour removal.

The results however will remain very valuable for comparison with any similar survey in the years to come

Author Response

Line 334 'concentration'

            Dear reviewer, thank you, we corrected into “concentrations”

It appears that you have made the very best use of the data available. The general use of antibiotics seems to be much more refined than was probably the case years ago. The veterinarians should be applauded in  restricting the use in most categories. Interpretation of the results was unfortunately limited by the data limitations. It would have been valuable to know why and under what circumstances the more valuable antibiotics were used. Also although the data was arranged into different categories, they were too general to examine the use of different antibiotics in specific disease areas. For instance prophilactic treatment after major abdominal surgery compared with a cutaneous tumour removal.

            Dear reviewer, thank you for your comment, and we agree that further studies hopefully can analyse the data more specifically. We understood your text as a comment, but not as a request to revise the text. Please let us know if you have any change requests.  

The results however will remain very valuable for comparison with any similar survey in the years to come.

            Thank you very much for your comments.